# Optical Sensor for Monitoring Leakage Current and Weather Conditions in a 500-kV Transmission Line

**DOI:** 10.3390/s22135034

**Published:** 2022-07-04

**Authors:** Marcelo M. Werneck, Paulo Henrique S. Pinto, Renato T. Bellini, Regina Célia S. B. Allil

**Affiliations:** Photonics and Instrumentation Laboratory, Universidade Federal do Rio de Janeiro, Rio de Janeiro 21941-901, Brazil; paulo.hsp11@gmail.com (P.H.S.P.); rtbellini@gmail.com (R.T.B.); reginaallil@coppe.ufrj.br (R.C.S.B.A.)

**Keywords:** leakage currents, plastic optical fiber, optical fiber sensor, high-voltage transmission line, polluted insulator

## Abstract

The leakage current (LC) caused by the surface contamination of insulators, together with environmental variables, is one of the most basic online monitoring parameters for insulator status. However, the impact of weather conditions such as temperature, air humidity, and dew point on the LC has not been deeply studied until now. In this paper, based on meteorological data obtained online and LC obtained with an optical fiber sensor, installed in 500-kV insulator strings of a transmission line, the impact of weather conditions was studied. Results indicate that the LCs follow a specific pattern, according to weather conditions. The system has been continuously monitoring LC, humidity, temperature, and dew point uninterrupted for three years, sending the acquired data to a web page; therefore, it has been demonstrated to be robust, reliable, and repetitive. The sensor features the broadband response and acquisition capabilities of partial discharge pulses in high-voltage insulators, allowing the detection of high-frequency pulses. When comparing the LC measured in this work with those from other works, our measurements are substantially higher; this is due to the type of pollution found in this specific situation, which includes iron oxide powder, producing a conductive layer over the insulator surface that, unlike sea salt, does not depend on humidity to conduct an LC. One of the conclusions reached in this work is that partial discharge surges are caused when the local temperature reaches the dew point and not simply from the presence of high humidity, as stated in many works dealing with LCs. The monitored LC can be used as an indicative parameter of a possible flashover, enabling the proper planning of insulator predictive maintenance, either by jet-washing the surface or even changing the insulators when they are damaged.

## 1. Introduction

Insulators in high voltage transmission lines (TL) are key components in energy distribution systems, remaining in the field for long periods. As is widely known, they are used to hold conductors in position, separating them from one another and from the surrounding structures. They are also expected to resist mechanical stresses, to resist electrical stresses (over-voltages), and resist environmental stresses. But although they are supposed to be perfectly non-conducting devices, their performance is compromised by environmental pollution.

Environmental pollution, depending on the location of the TL, can vary from dust, smoke, clay powder, or chemicals from nearby industries to salt spray, which progressively coats the insulator surface with a conductive deposit, leading to leakage currents (LC).

The most common pollutant is the salt coming from the sea that is carried by the wind. The salt is an insulator, but it conducts electricity when wetted by fine rain or by the night dew, causing the surface deposit to conduct a current from the high-voltage side of the insulator to the ground side through the insulator’s surface.

When a leakage current flows along a path over the wet polluted surface of the insulator, the heating effect of this current dries out one or more narrow bands, known as high-resistance dry bands, that interrupt the currents and, in sequence, carry virtually all the voltage applied to the insulator. Discharges across these dry bands can grow and span the complete length of the insulator, ionizing the surrounding air and making it conductive, producing a flashover of the insulator. The flashover is an electric discharge over and around the surface of the insulator.

LC is formed by a summation of the partial discharge (PD) that is caused by deterioration in the insulation of high-voltage systems. It is a progressive degradation that often starts with a small current but finally leads to the destruction of the insulator when a flashover occurs. Once the PDs occur, the lifespan duration and performance of the insulators will degrade. Without any treatment of the insulators, the discharge current will eventually bridge the two sides of the insulator completely, which results in a large short-circuit current, known as a flashover.

According to Katsuse et al. [1], partial discharges are of extremely short duration, about tens of nanoseconds. Broadband sensors with higher sampling rates are, thus, required.

Flashovers normally cause power blackouts, leading to fines being applied to the TL operator company by the national grid controller and losses due to the lack of energy production. To circumvent the likelihood of flashover, operators can increase the creepage length of the insulators by adding extra sheds, but this is an expensive procedure that increases the costs of the installation and, ultimately, would not resolve the problem.

Alternatively, the TL operators can contract a special service that uses a water jet to periodically wash the insulators. The normal periodicity is once a year, but this could be amended if the extent of the pollution deposit is known.

Among the different insulator-monitoring techniques, the equivalent salt deposit density (ESDD), described in the IEC 60815-1 [2], stands as the best approach to monitoring the extent of the conductive pollutants that can accumulate on the insulator’s surface [3]. It provides a conventional number, based on the amount of conductive deposition, leading to a reduction in dielectric strength and surface resistivity. It is simple to evaluate and requires only demineralized water to wash a test insulator and a conductivity meter; however, the transmission line operators need to maintain a trained team to visit the strategic points of the transmission line periodically, to collect the ESDD information.

According to the IEC 60815-1, using the ESDD in mg/cm^2^, five classes of pollution site severity are quantitatively defined, from very light pollution (ESDD < 0.01 mg/cm^2^) to very heavy pollution (ESDD > 0.40 mg/cm^2^).

Another method of monitoring insulator performance is by measuring the leakage current, which, if properly interpreted, can provide a measure of how close the insulator string is to flashover. A great deal of research has been carried out concerning the effect of humidity on leakage in insulators, and the next section will describe some of these techniques.

For this work, an optical fiber sensor has been developed that measures the LC of 500-kV string insulators in three different locations of a transmission line, together with information on temperature, humidity, and dew point. These four parameters are transmitted in real time via a cell phone network (GPRS) to a server that can be accessed by the internet. Before describing the developed system in detail, in the next section, we describe the history of other works on LC measurement, comparing them with the advantages and innovations that the present work offers in terms of flashover monitoring and prevention.

## 2. On Measuring Leakage Currents on Insulators

Leakage currents are known to be the primary cause of flashover in insulators and have been measured for decades. We can mention, for instance, the work of Karady and Amarh in 1999 [4], which used four porcelain insulators energized at 50 kV in an artificially contaminated chamber. The LC was measured by a 130 Ω shunt resistor and recorded by an analog magnetic tape recorder with 5 kHz bandwidth. Although they used too narrow a bandwidth to be able to observe the high frequency of the partial discharge pulses presented in LC waveforms, the idea was to study the LC waveforms just before the flashover occurred. For this kind of measurement, the insulator must be separated from the ground potential by the shunt, a procedure that would be impossible to perform in a real insulator.

Another method to measure the LC was proposed by Chen et al. [5], who developed a broadband transducer based on a Rogowski coil, designed for the online monitoring of partial discharge current waveforms. Using the designed transducer, they monitored leakage current and the partial discharge signals of 10-kV polluted synthetic insulators in a climate chamber. To extract the partial-discharge signal from the LC, a broadband transducer is required, and its low-cutoff frequency should be below the power frequency. However, in this case, the ring transducer must also be connected around a conductor that connects the insulator to the ground potential; obviously, this is impossible to utilize in a real situation.

Our team has been working on this subject since 2015 when a sensor was developed using a plastic optical fiber (POF) and an LED to measure the LC, which was capable of being applied directly to the insulator string without interfering with the local installation [6]. The idea at the time was to make a proof-of-concept in the use of POF-based sensors for LC measurement. The sensor was installed on the tower, close to the end of the insulator string, and a short electric cable detoured the LC from the last shed of the insulator string to the sensor. The sensor converted the current on an amplitude-modulated light that is conducted by the POF to the transmitter installed at the foot of the tower. The application of this technique can also be seen in the works of Fontana et al. (2006) [7], Wang et al. (2012) [8], and Yan Mi et al. (2014) [9].

Several other authors have also studied this effect using different techniques and for various purposes, which we describe below.

The technique of bypassing the LC from the last shed of the insulator string was extrapolated from the work of Fierro-Chavez et al. [10], who were the first to directly implement this technique on glass insulator strings to diverge and measure leakage currents. They used a current transformer where the primary was directly connected between the first and the second sheds of the string (ground side).

Other methods to measure LC using optical fiber can be seen in the literature, such as in the work of Yan et al. [11], where they applied an FBG fixed to a cantilever that is attracted by a magnetic coil in which the LC flows. Optomechanical methods involving moving parts can work well in the laboratory environment, but they would not be practical in the field.

The fact that the air humidity is the sine qua non condition to start a flashover process in polluted insulators has been known for decades; a great deal of research has been carried out concerning the effect of humidity on the moisture level of insulators, but it is only recently that works on this subject have appeared in the literature. A good example is the work of Meyer et al. [12], in which an LC monitor was developed and installed in the field to monitor leakage current, ambient humidity, and temperature. They used a shunt and measured currents up to 2 mA but could not show a clear relationship between humidity and LC; however, it became clear that the peaks of LC only occurred with a humidity level higher than 80%.

In laboratory tests using an artificial climate chamber, Wang et al. [13] concluded that when polluted insulators get wet, the necessary voltage to cause a flashover decreases. The results indicate that the water absorption of the polluted layer is proportional to the temperature difference between the ambient and insulator surfaces.

More recently, Roman et al. [14], when correlating the LC with temperature, rain, and humidity, concluded that LC increases when the humidity rises to 90% and the temperature drops to about 5 °C.

Based on the ambiguities between the different works mentioned above, their characteristics, and their respective limitations, we designed our system for monitoring LC and weather parameters to be installed on a 500-kV TL, which will be detailed in the next section. This work is expected to encompass all the information available on weather conditions, to compare the results and explain the leakage current behavior.

## 3. Materials and Methods

A 5-km, 500-kV TL that connects the thermal power plant of a steel company to a national grid substation registered a flashover causing a crack in one of the insulators. The company contracted our laboratory to design, develop, and install a device to measure the LC in three different locations, along with the TL. The developed system aims to measure, in real time, the temperature, air humidity, and LC, transmitting these values along with the calculated dew point via GPRS (General Packet Radio Service) to the computer server of our laboratory. The main idea was to study the correlation between weather parameters, leakage currents, and the occurrence of flashovers. The obtained values are being used to study pollution on the insulators, indicating the optimum time for changing or washing them.

To measure the leakage currents of insulators in a live transmission line, the first concern is that whatever the chosen technique, it should not interfere with the insulators and the tower’s performance. One of the best methods for monitoring high-voltage parameters is utilizing optical fiber technology, which has been used for many years in high-voltage applications. Additionally, plastic optical fiber technology presents many advantages over silica fibers, among which, one can mention:Easy manipulation and connecting, due to its larger diameter (1 mm) compared to silica fibers;The small bending losses due to its robust guiding capability, with high numerical aperture and super-multimode characteristics;Passive sensing technology does not need electrical energy at the sensing point;Electric insulation allows measuring to be directly over the high-voltage potential;The tools, devices, and accessories required for POF are much cheaper than their counterparts for silica fibers.

The only disadvantage of POF, compared with silica fiber, is its high light attenuation of 70 dB/km, compared with 0.25 dB/km for silica fiber. This high attenuation compromises long-distance telemetry but is not relevant in our case, where there is only 50 m between the sensing point and the receiver.

The sensing principle selected for this work is based on the ability of a high-brightness LED to provide an output light that is proportional to the current passing through it. The LED color wavelength chosen for this work provides the best temperature-independent output light, compromised with the optical fiber’s smallest attenuation window, 70 dB/km at 565 nm. For guiding the LED light to the receptor for the ground potential, a 50-m POF cable was used, which presents a total attenuation of 3.5 dB at the chosen transmission window.

The main part of the system is the current transducer, which comprises an LED-based sensor installed on the insulator, a POF cable (Mitsubishi Rayon Co., Tokyo, Japan), and an optoelectronic receiver, installed at the tower foot. Figure 1a shows the block diagram of the transducer. The LC is collected from the last shed of the insulator string, crosses the sensor, and returns to ground potential at the tower steel structure.

Fierro-Chavez et al. [10] were the first to suggest this technique, directly implemented on glass insulators string, diverging the leakage current to the sensor to measure it. They used a current transformer where the primary was directly connected between the first and the second shed of the string (ground side).

In this work, for the current sensor, we used a high-efficiency green LED (peak wavelength of 565 nm) after a full-wave rectifier, as shown in Figure 1a. The electronic system is protected from high voltage transients by a bidirectional transient voltage suppression diode. In this way, the current pulses are converted to amplitude-modulated light pulses that are guided by the POF to the receiver, the optical sensor module (OSM). The OSM is a printed circuit board with high-bandwidth components, such as a trans-impedance amplifier to convert the photodetector reverse current to voltage and a DC to RMS converter with a controlled integration time. Therefore, V_out1_ is similar to the real LC crossing the insulator. V_out2_ is the root mean square (RMS) of V_out1_, integrated over a few milliseconds to circumvent the limitations of the low bandwidth and low-sampling frequency of the commercial controller, which is responsible for data transmission via GPRS inside the remote unit. V_out2_ is also used for calibration purposes.

Figure 1a shows the block diagram of the leakage current sensor and the optical sensor module. The electronics are protected inside a ceramic cup immersed in resin, to provide an IP66-rated enclosure (Figure 1b). The POF cable and the electric electrode that will be connected to the insulator are also shown. Figure 1c shows the OSM circuit board of the optical receiver, which is installed inside the RU.

The sensor was calibrated by injecting a variable DC current in its input and monitoring V_out2_, to obtain a relationship that is saved in the controller’s memory. For testing the current sensor behavior under high-frequency current pulses, a variable frequency sinusoidal current provided input to the sensor, and its outputs V_out1_ and V_out2_ were monitored.

Figure 2 shows the block diagram of the remote unit (RU). It comprises three sensors (leakage current, temperature, and air humidity), a programmable logic controller (PLC), and a router. The PLC, run by software written in LADDER, reads the three sensors, calculates the dew point, and transmits these data via a local serial bus to the router containing a cell phone chip, allowing communication with the internet via GPRS. The PLC, the router, the temperature and humidity sensors, and the power supply are commercially available from Novus Automation, Inc., Canoas, Brazil. The system is energized by a solar panel and an inverter. The controller reads the sensors every 30 s and stores the data via comma-separated values (CSV) in its internal memory, to send the bulk data once a day to the laboratory server.

The dew point is calculated by the PLC from the humidity and air temperature, informed by their respective sensors. Equation (1) calculates the dew point:(1)$$DP=T-\frac{\left(100-H\right)}{5}$$
where *DP* is the Dew Point in °C; *T* is the temperature in °C, and *H* is the relative humidity as a percentage. This simple approximation for the dew point is accurate to within about ±1 °C for relative humidity above 50%.

After the calibration test, the sensor system was tested in a salt fog chamber, according to the international standard, IEC 60,507 [15]. The salt fog method uses a salt solution of sodium chloride of commercial purity, kaolin, and tap water. The fog is produced in the test chamber by employing a spraying system that atomizes the solution with a stream of compressed air. This test is intended to replicate the conditions at the seashore in which the insulator is energized and is exposed to salt fog for some time.

Figure 3 shows the block diagram of the test. The variac transformer, together with the high-voltage transformer, generates a voltage of up to 35 kV that is applied to the insulator under test. At this point, the voltage and current can be monitored by the oscilloscope through a high-frequency current transformer (HFCT) and a high-voltage probe. The LC and PD flowing on the insulator are directed to the sensor and then to the ground potential through a 10-Ω shunt resistor. The voltage over the shunt resistor, divided by 100, is the total LC that can be seen on the oscilloscope. Finally, the output of the OSM is monitored by the oscilloscope, where the real and measured currents can be compared.

The next test is the whole transmission chain, starting at the current sensor, optical fiber transmission, OSM, PLC, router, GPRS transmission, the internet link, and the data arriving via CSV at the laboratory server. For this test, the block diagram shown in Figure 4 is used. The high-voltage process is the same as the one shown in Figure 3, except that now the OSM lies inside the remote unit that transmits the data directly to the internet. The data arriving at the laboratory server are monitored throughout the test and are checked for continuity and consistency.

## 4. Field Installation

With all laboratory tests performed, three units were taken to the field for installation on three 500-kV transmission line towers. This TL transports the energy generated by the Ternium Steel Co. thermoelectric plant to the Furnas National Grid Substation, both located in Santa Cruz, Rio de Janeiro, Brazil. Figure 5 shows a satellite picture of the area, indicating the three towers where the sensors were installed and the steel company area, as well as the location of the iron ore storage stack and the southwest wind direction. This wind direction prevails during part of the year and blows salty sea fog from Sepetiba Bay, along with iron oxide dust from the company’s stack toward the TL, to accumulate on the surface of the insulator.

Figure 6a shows two live-line personnel climbing Tower 19, one on each side, and in Figure 6b, the technician is connecting the sensor (center-left of the picture) between the two last sheds of a 25-unit insulator string, consisting of typical glass discs. The remote units (RU) were installed on the foot of the tower, Figure 6c, and are energized by a photovoltaic system installed on the tower.

## 5. Results and Discussion

This section presents a discussion of the results achieved with the leakage current sensor from the tests in the laboratory, as well as data from the field.

### 5.1. Thermal Drift of Optical Sensor Module

For an optical fiber transmission system working using the amplitude modulation technique, our first concern is the amplitude variation due to the influence of secondary parameters, such as fiber curvature or temperature variations in the electronic parts of the system.

Bend losses mean that optical fibers exhibit additional propagation losses by coupling light from core modes (guided modes) to cladding modes when the fiber is bent. Typically, these losses rise very quickly once a certain critical bend radius is reached. However, this critical radius can be very small (a few millimeters) for fibers with robust guiding characteristics, such as fibers with a high numerical aperture. This is one of the advantages of the 1-millimeter-diameter plastic optical fiber, which contains millions of guided modes and is relatively immune to bend losses.

Temperature does not cause fiber losses; however, it does change the LED performance as well as the photodiode sensitivity. For checking these effects on the sensor system, the optical transmitter, a 50-m length of fiber optic cable, and the receptor were placed in an oven and the system performance was measured at temperatures from 25 °C to 65 °C. Figure 7 shows the result of such a test for the different leakage currents. It is possible to deduce that as the temperature increases, the LED output power decreases and the photodetector response decreases for any current. Bearing in mind that the average leakage current is around 20 mA, the uncertainty at this temperature range is about 3.8%, which is perfectly acceptable for the type of information that we require.

### 5.2. Sensor Calibration

The next test is the calibration of the whole system, which was conducted by injecting a variable DC current into the sensor terminals and monitoring V_out2_ (see Figure 1). This result is shown in Figure 8 and is saved into the PLC memory. They are separated into two ranges to ensure better sensitivity since the response of the system tends to saturate after 50 mA, but the LED supports up to 100 mA.

### 5.3. Sensor Frequency Response

According to Chen et al. [5], leakage currents are composed of several high-frequency discharges, known as partial discharges. In their work with artificially polluted insulators in a climate chamber with a series configuration of a current transducer, Chen et al. recorded pulses of up to 12 kHz, superimposed on a 60-Hz sinusoidal signal. In the present work, we applied high bandwidth devices in the whole measuring chain that were capable of detecting high-frequency PD pulses. For testing the frequency response of the current sensor, a variable-frequency sinusoidal current provided input to the sensor and its outputs V_out1_ and V_out2_ were monitored. Figure 9 shows the result of such tests. The input current was measured on a 10 Ω shunt resistor, connected in series with the sensor.

The frequency response curve shown in Figure 9a demonstrates the circuit’s capability of responding to frequencies that are close to 1 MHz. Indeed, Figure 9b shows the circuit response to an input frequency of 500 kHz; therefore, it is capable of detecting PDs at high frequencies.

### 5.4. Tests in the Fog Chamber

The test of the system when under leakage currents was performed according to the block diagram shown in Figure 3. While the test insulator is polluted by the stream of salt fog, the input voltage is increased until partial discharges start to show. Then, oscilloscope captures are taken, showing the leakage current measured at the shunt resistor, the current waveform as identified by the system, and the equivalent RMS current that will be read, stored and transmitted by the remote unit. The goal of this test is to compare the real LC with the recorded one, as identified by the oscilloscope at V_out1_, the peak-to-peak response of the OSM, and V_out2_, the RMS equivalent of V_out1._

Figure 10 shows a typical PD pulse, as seen at the shunt (upper trace) and recovered by the OSM (lower traces). It is composed of a short burst of 0.5 ns duration, superimposed with an oscillation of about 10 MHz. This result demonstrates the system’s capacity to detect high-frequency pulses; although the 3 dB cutoff frequency was much smaller (see Figure 9a), the transducer was capable of detecting high-frequency pulses.

Figure 11 shows a typical PD repeatability, with 10-MHz short bursts at a repetition frequency of about 250 kHz. As mentioned before, the analog to digital converter (ADC) of the commercial PLC used in the RU cannot acquire the high frequencies produced by the PD and LC; for this reason, we included an RMS converter on the OSM circuit with a programmable integration time. In Figure 11, the lower trace is the RMS equivalent of the LC, as identified by the OSM, which can easily be converted by the ADC of the PLC.

Figure 12 shows the same scenario as in Figure 11, but with a lower horizontal scan frequency. It is possible to deduce that the RMS equivalent of the LC follows the PD bursts in amplitude, showing higher amplitude when the bursts increase in power. Similarly, as seen in Figure 13, for wider PD bursts, the RMS also shows wider pulses. In the long run, the RMS pulses present a repetition rate of about 60 Hz and are, therefore, capable of being acquired by the PLC.

The 60 Hz repetition rate of the bursts shown in Figure 12 and Figure 13 also appears in the work of Chen et al. [5]. Basically, the PDs occur at the top of the high-voltage sinusoidal waveform and are either positive or negative since the zero-crossing periods do not present enough voltage to generate PDs. In their work, pulses of 12 kHz were seen superimposing the peaks of the 60 Hz sinusoidal waveform. These peaks are equivalent to the bursts we see in Figure 11, which were converted into simple peaks due to the low-frequency response of their system.

### 5.5. Test of GPRS Transmission Capability

This test was performed according to the block diagram shown in Figure 4, to test the transmission capability of the RU. The protocol was essentially the same as that used for the tests performed in the previous section, except that the oscilloscope was used to check the measurements. Here, the system acquires the leakage currents, together with the local weather parameters, compacts these data into a CSV format, and sends them via GPRS to the laboratory server in that format. Figure 14 shows the data acquired and transmitted over a period of 1 h 20 min. During this time, we maintained a fog stream in the test chamber and a working voltage of 20 kV.

With this test, we assured the system transmission capability; therefore, it was then ready to be taken into the field.

### 5.6. Field Results

Three leakage current systems were installed on the 500-kV transmission line, according to the satellite picture shown in Figure 5. The objective of this choice was to completely monitor the entire LT, since the first tower, which is closer to the sea and the iron ore stack, might show a higher pollution accumulation than the other towers. However, we did not observe different degrees of pollution when comparing data from different towers, which confirmed that the LT was relatively short and completely encompassed by the local micro-climate. In this section, we show the different situations combining leakage currents and weather conditions. The system has already been installed in the field for three years, producing a huge amount of data but, in this paper, we have only presented the days where increased LC activity has been observed, combined with the weather parameters. In all graphs in the figures below, the orange line is the humidity, the blue line is the temperature, the gray line is the dew point, and the yellow line is the leakage current.

Figure 15 shows favorable weather conditions in which the insulator remains at an average leakage current below 15 mA. Although the humidity rises to almost 90%, no increase in leakage current was observed, confirming that in dry conditions, no LC occurs. Roman et al. [14] concluded in their work that the LC increases from 0.01 mA to 0.08 mA when the humidity rises to up to 90% and the temperature drops to about 5 °C. By inputting these weather conditions in Equation (1), we find a dew point of 3 °C, which is below the local temperature; therefore, no humidity would accumulate on the insulator surface. This explains the low LC. In the present work, the humidity also increased but the local temperature remained above the dew point, which guaranteed a constant LC.

Notice that the leakage currents measured here are well above those reported by Meyer et al. [12] and Roman et al. [14]. This is because these insulators are polluted not only by NaCl, which is soluble and susceptible to humidity, but also by non-soluble iron oxides, such as Fe_2_O_3_, which are conductive even without humidity. This effect is discussed by Zhang et al. [16], who suggest that small amounts of Al_2_O_3_ and Fe_2_O_3_ on the insulator’s surface can cause the flashover voltage to decrease by 20%.

Figure 16 shows a sudden increase in humidity above 90% at midnight, together with a temperature drop that is close to reaching the dew point. The consequent increase in the insulator surface humidity has made the LC rise to 15 mA. However, just after dawn at 8 a.m., the temperature rose to 38 °C and the humidity dropped to 50%, with the consequent dropping of the LC to 10 mA.

Figure 17 shows some of the worst conditions that a polluted insulator can experience: high humidity, cool weather, and heavy rain, but these conditions can warn the operators that a flashover is imminent. In this specific case, the dew point was reached by 7 p.m. and partial discharges started to show. PDs, when triggered, generate dry bands; that is, during the arc propagation process, the path that the discharge follows over the insulator is dried out due to the generated heat, stopping the current. Normally, new paths can occur intermittently, which is the scenario shown in Figure 16. However, if there is heavy rain, all possible paths are open again after drying, and the leakage current increases, presenting a situation in which LC peaks or even a flashover can occur. In the case shown in Figure 17, there was heavy rain by 8 p.m., then there was a thunderstorm from 9 p.m. to 11 p.m., followed by light rain and mist. These conditions triggered the LC peaks to rise to 35 mA, as seen on the graph. Notice that a current of just 35 mA under 500 kV generates a heat of 17.5 kW. This heat dissipates quickly; however, many discharges occurring within a short period can damage the insulator.

The weather conditions were confirmed by the website of the Meteorological Station of Santa Cruz Air Base, located about 6 km south of the transmission line. The following conditions were reported on 31 March 2022 [17]:20:00 h: Heavy Rain, Mist;21:00 h: Thunderstorm with Light Rain;22:00 h: Thunderstorm with Light Rain and Mist;23:00 h: Recent Thunderstorm, Rain, and Mist.

Figure 18 shows another situation in which a sudden thunderstorm by 8 p.m. increased the humidity above 90% and dropped the temperature, which then reached the dew point. This situation immediately triggered a peak in LC. By 9 p.m., the thunderstorm changed to light rain that did not support LC peaks but instead kept the LC to 20 mA. By 6 a.m., the temperature rose with the dawn; the humidity dropped, and the leakage current decreased to 15 mA.

The weather conditions were confirmed using the website of the Meteorological Station of Santa Cruz Air Base. The following conditions were reported on 26 March 2022 by the Meteorological Station at Santa Cruz Air Base [18]:20:00 h: Thunderstorm;21:00 h: Recent Thunderstorm, Light Rain;22:00 h: Mist.

## 6. Conclusions

A fiber optic-based leakage current monitoring system was developed and installed in three insulators, along a 500-kV transmission line.

This paper, based on previous research, presents improvements such as a broadband receiver, a robust and reliable sensing and transmitting system, and the inclusion of dew point information. These improvements made it possible to detect high-frequency partial discharges, study the behavior of polluted insulators over a long period, and correlate the climate parameters with the occurrence of leakage currents.

The main conclusions taken from the results include:The system has continuously monitored leakage current, humidity, temperature, and dew point uninterrupted for three years and has been sending data to a web page, thus, demonstrating that it is robust, reliable, and repetitive.For the first time, a robust and reliable TRL 8 device has been installed on 500-kV towers to monitor and long-distance transmission insulator’s leakage current and weather parameters.The sensor features the broadband response and acquisition capabilities of partial discharge pulses in high-voltage insulators, allowing the detection of high-frequency pulses.When comparing the LC measured in this work with those from other works, our measurements are substantially higher. This is due to the type of pollution found in this specific situation that includes iron oxide powder, producing a conductive layer over the insulator surface that, unlike sea salt, does not depend on humidity to conduct an LC.Partial discharge surges are caused when the local temperature reaches the dew point rather than simply because of high humidity, as stated in many works dealing with leakage currents.

The insulator leakage current monitoring can be used as an indicative parameter of a possible flashover, enabling the proper planning of predictive insulator maintenance, either by jet-washing them or even changing them when damaged. However, for this technique to be of any use to the electric company, it is necessary to transform the collected data into information. This means establishing parameters that can determine the actual status of the insulator regarding the leakage current that flows to the ground. A good way to establish these parameters is by correlating the average leakage current with the ESDD taken periodically from each tower. When this correlation is consistently established, the ESDD can be interrupted, and the technicians can use the LC as a monitoring parameter.

## Figures and Tables

**Figure 1 sensors-22-05034-f001:**
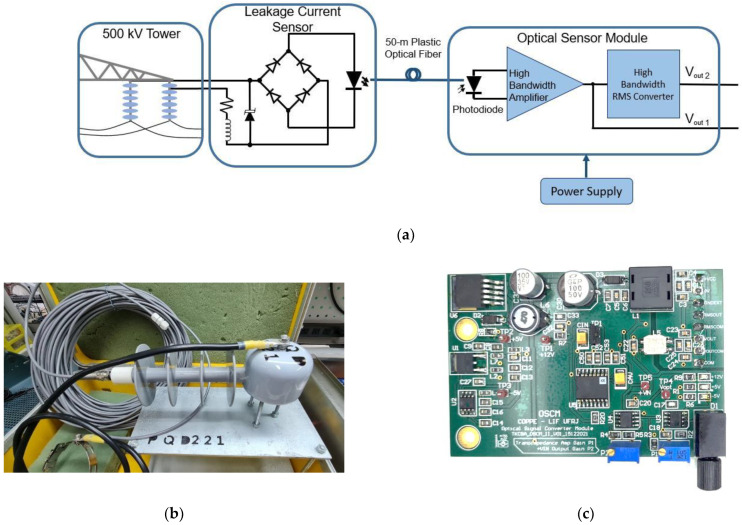
(**a**) Block diagram of the leakage current sensor and optical sensor module; (**b**) a sensor inside an IP66-rated enclosure, with the POF (plastic optical fiber) cable and the electric electrode; (**c**) PCB of the optical receiver.

**Figure 2 sensors-22-05034-f002:**
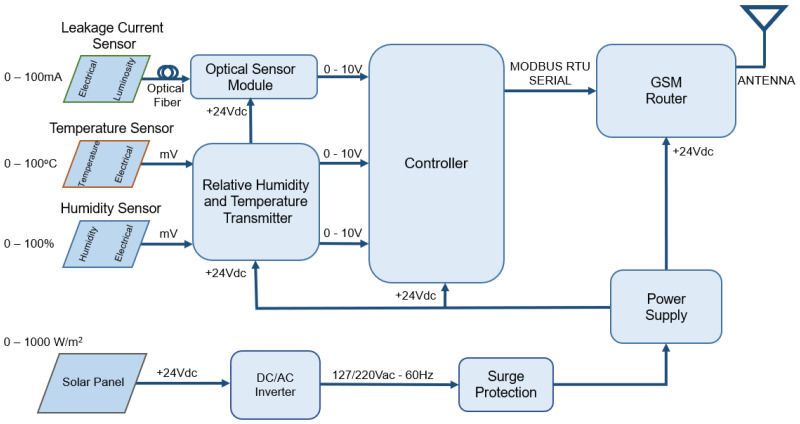
Block diagram of the Remote Unit.

**Figure 3 sensors-22-05034-f003:**
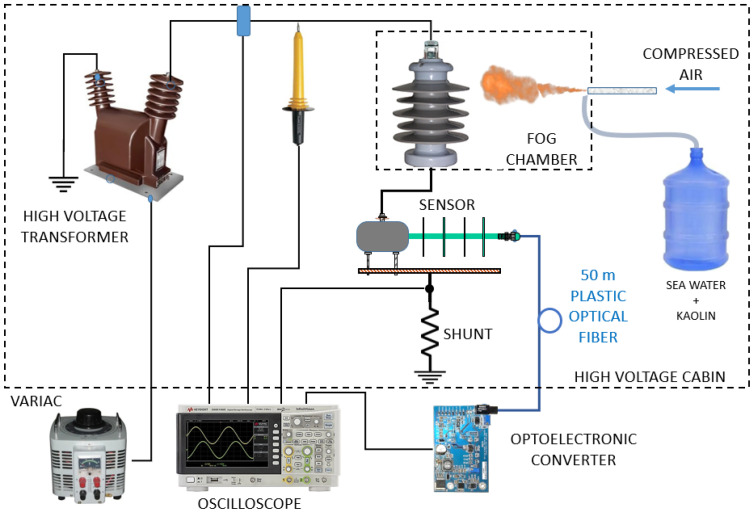
Block diagram of the high-voltage test chamber.

**Figure 4 sensors-22-05034-f004:**
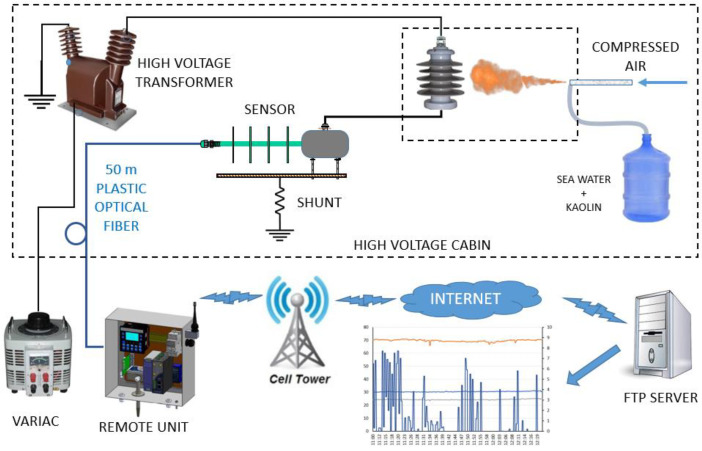
Block diagram of the transmission chain, starting at the current sensor, optical fiber cable transmission, the OSM, PLC, and router inside the remote unit, the GPRS transmission, the internet, and the data arriving via CSV at the laboratory server.

**Figure 5 sensors-22-05034-f005:**
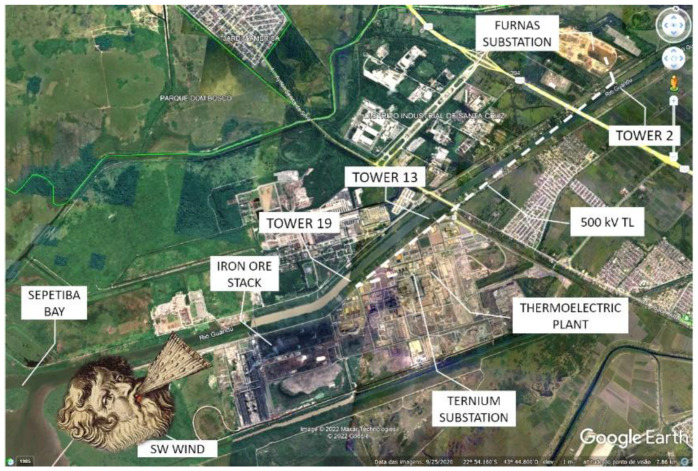
The transmission line and the three towers, with the sensors installed. The transmission line transports the energy generated by the Ternium Steel Co. thermoelectric plant to the Furnas National Grid Substation.

**Figure 6 sensors-22-05034-f006:**
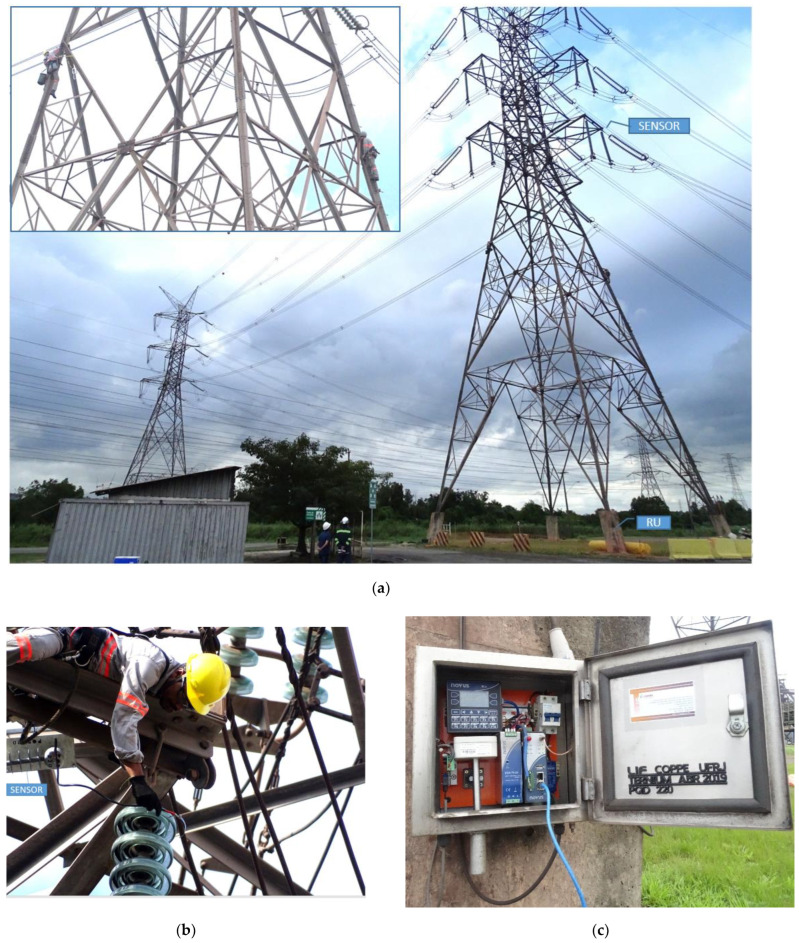
(**a**) Live-line personnel climbing Tower 19 for the installation of the sensor; (**b**) The technician is connecting the sensor between the two last sheds of a 25-unit insulator string; (**c**) The remote unit (RU), fixed at the foot of the tower.

**Figure 7 sensors-22-05034-f007:**
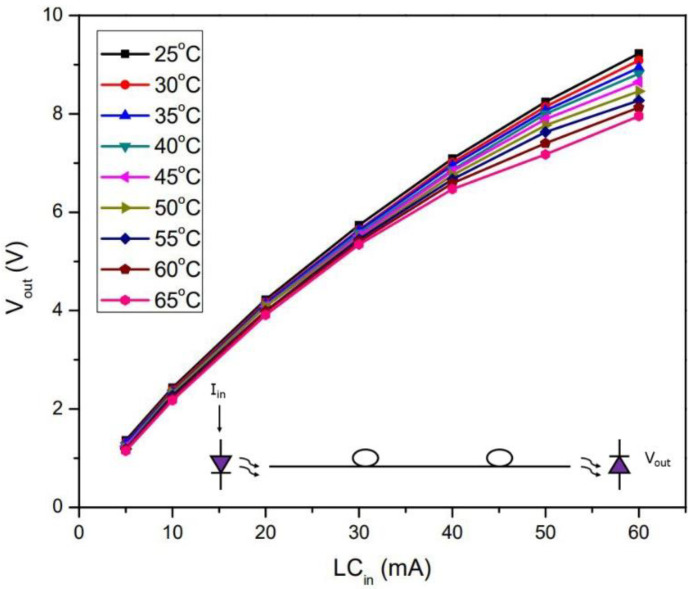
The temperature drift of the sensor from 25 °C to 65 °C, considering the input current on the green LED, varying from 0 to 60 mA, and using a 50-m length of POF cable.

**Figure 8 sensors-22-05034-f008:**
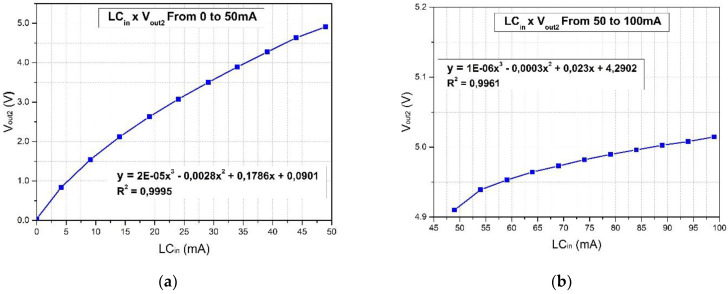
The calibration procedure of the sensor consists of the input of a current to the sensor and measurement of the output voltage at point V_out2_ (see Figure 1). (**a**) Calibration curve for an input range of between 0 and 50 mA; (**b**) calibration curve for an input range of between 50 and 100 mA.

**Figure 9 sensors-22-05034-f009:**
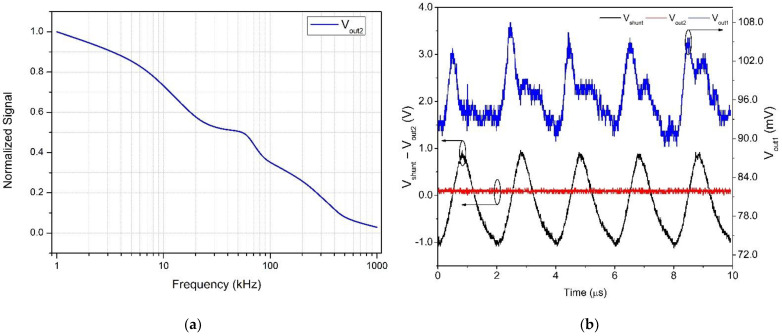
System response of the sensor for an input sinusoidal waveform at different frequencies. (**a**) Frequency response curve, demonstrating the circuit’s capability of responding to frequencies close to 1 MHz. (**b**) Black trace: a 500 kHz sinusoidal input current, measured on a 10 Ω shunt resistor. Blue trace: response of the system to a 500 kHz input (V_out1_, Figure 1a). Red trace: RMS equivalent of the input current (V_out2_) (Black circles around curves indicate the applicable vertical axis).

**Figure 10 sensors-22-05034-f010:**
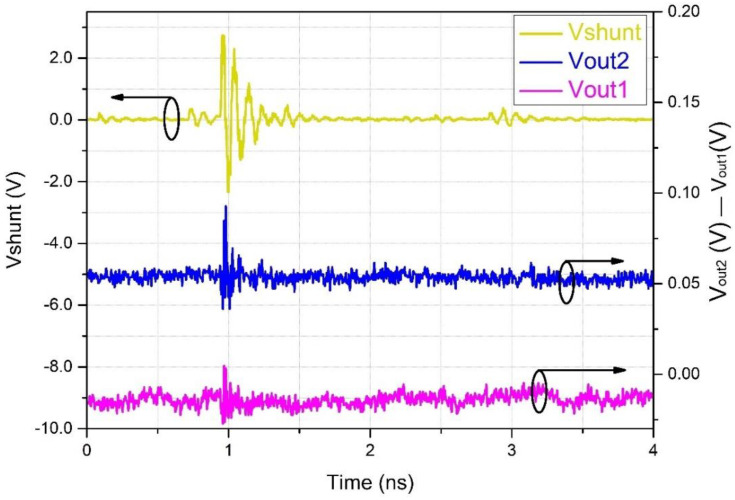
Partial discharge burst, with a frequency of about 10 MHz. Upper trace: PD signal, captured by the 10-Ω shunt resistor. Center trace: signal captured by the OSM before the RMS converter. Bottom trace: signal after RMS conversion (black circles around curves indicate the related vertical axis).

**Figure 11 sensors-22-05034-f011:**
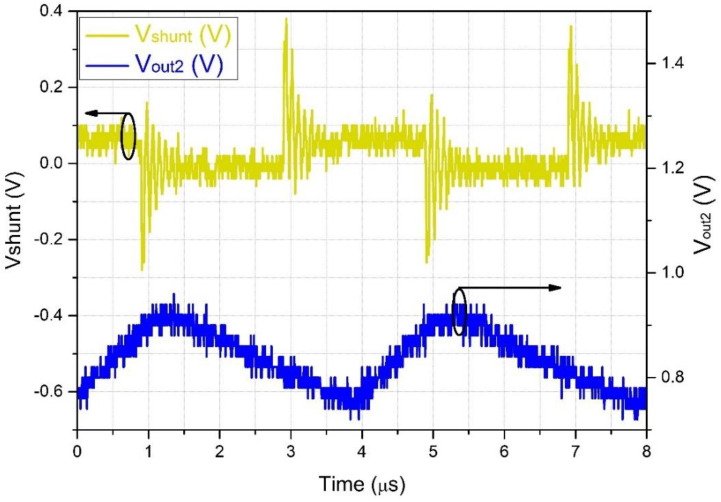
Partial discharge pulses with a frequency of about 10 MHz, with a repetition rate of about 250 kHz. Upper line: signal captured by the shunt resistor. Bottom line: signal captured by the OSM after the RMS converter (black circles around curves indicate the related vertical axis).

**Figure 12 sensors-22-05034-f012:**
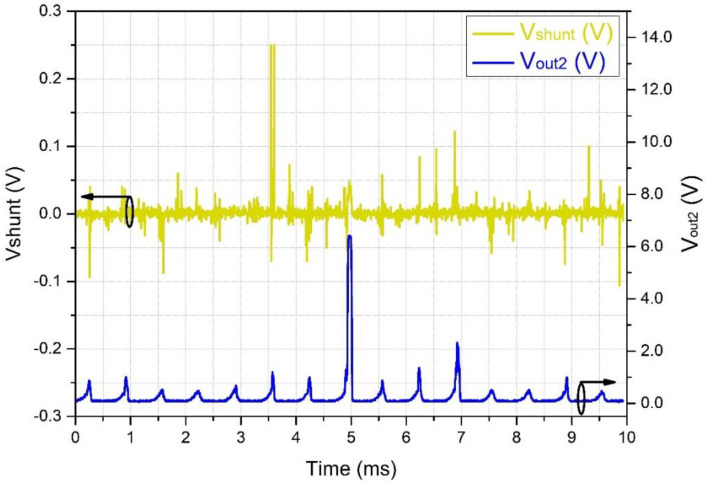
Partial discharge pulses of the leakage current. Upper line: signal captured by the shunt resistor. Bottom line: signal captured by the OSM after the RMS converter, showing higher amplitude when the bursts increase in power (black circles around curves indicate the related vertical axis).

**Figure 13 sensors-22-05034-f013:**
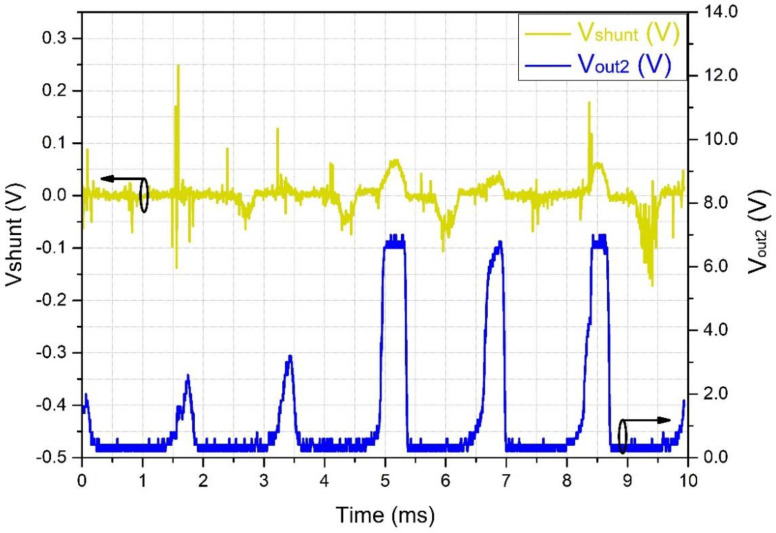
Effect of the RMS converter on partial discharge pulses. Upper line: signal captured by the shunt resistor. Bottom line: signal captured by the OSM after the RMS converter, showing wider pulses when the bursts increase in length (black circles around curves indicate the related vertical axis).

**Figure 14 sensors-22-05034-f014:**
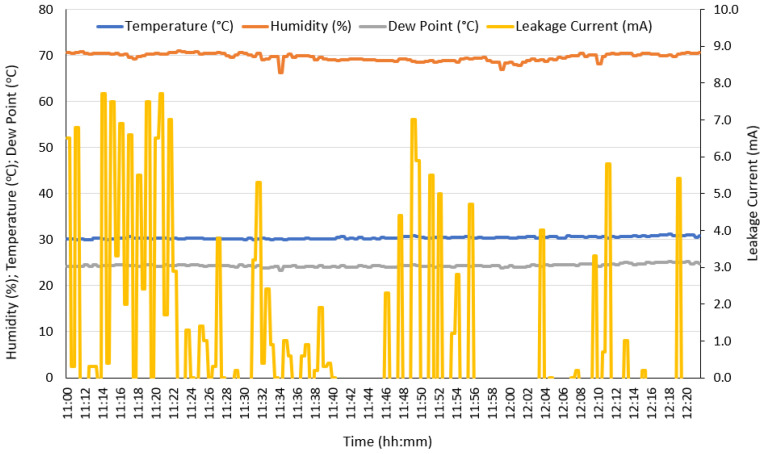
The data acquired in the test chamber and transmitted via GPRS to the laboratory server over a period of 1 h 20 min.

**Figure 15 sensors-22-05034-f015:**
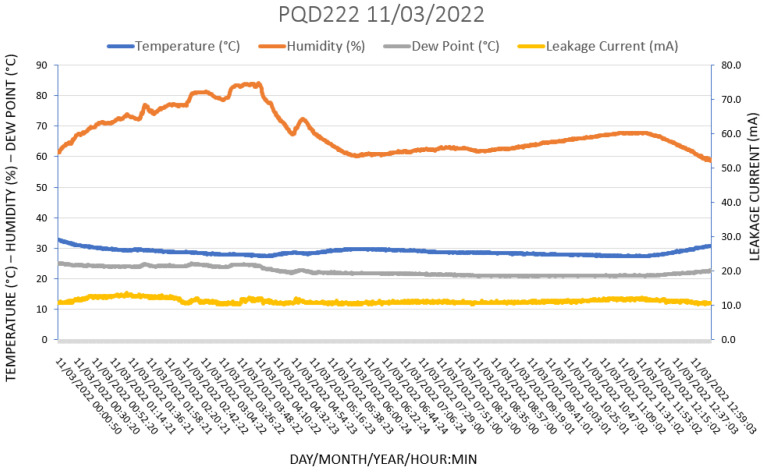
Favorable weather conditions in which the insulator remains at an average leakage current below 15 mA but no LC peaks are recorded, in particular, because the dew point is well below the local temperature.

**Figure 16 sensors-22-05034-f016:**
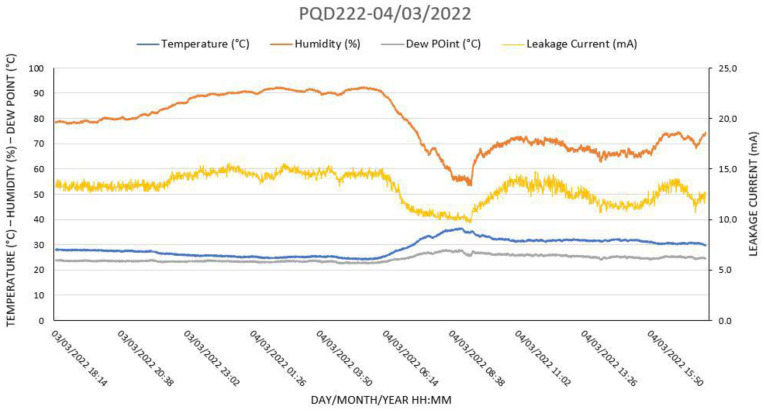
During the night, the humidity rose above 90%, together with a temperature drop that was close to reaching the dew point, producing a small rise of the LC to 15 mA. Note that just after dawn, the temperature rose to 38 °C and the humidity decreased to 50%, dropping the LC to 10 mA.

**Figure 17 sensors-22-05034-f017:**
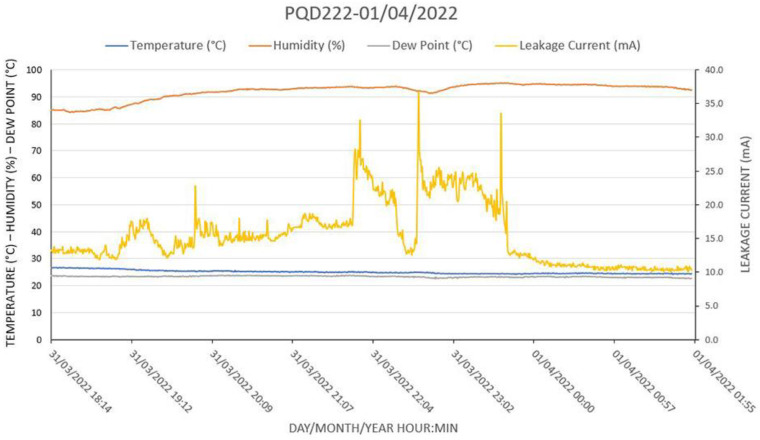
Some of the worse conditions that a polluted insulator can experience: high humidity, cool weather, heavy rain, and the temperature dropping to the dew point. The system detected an LC peak of 35 mA.

**Figure 18 sensors-22-05034-f018:**
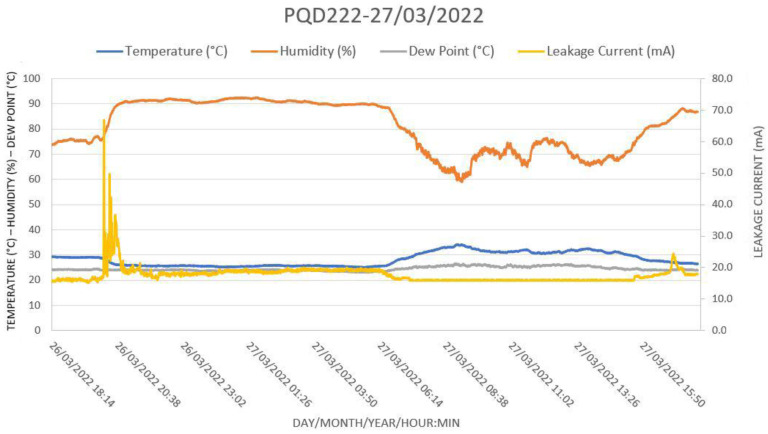
A sudden thunderstorm by 8 p.m. increased the humidity above 90% and dropped the temperature, which reached the dew point, triggering a peak of LC to 70 mA. By 9 p.m., the thunderstorm had changed to light rain, keeping the LC at 20 mA. By 6 a.m., at dawn, the temperature rose, the humidity dropped, and the leakage current decreased to 15 mA.

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
