# Peer review of "Optical Sensor for Monitoring Leakage Current and Weather Conditions in a 500-kV Transmission Line"

_sensors, 2022, doi:10.3390/s22135034_

Round 1

Reviewer 1 Report

The present work by Werneck et al. studies the plastic optical fiber based sensor for detecting and efficiently monitoring the inevitable leakage current in a 500-kV transmission system for various weather conditions conducted in Brazil country. The results were sound and of interest to a wide range of readers in this group. Since the present investigation carries enough scientific information to get published in this reputed journal, I recommend it for the same in the same version.

Author Response

Thank you for your support in this important research for the electric power sector.

Reviewer 2 Report

In this manuscript, a  POF sensor is used for detecting and monitoring leakage current in a 500-kV transmission line along with weather conditions. The field test results have important reference significance. However, there are very few descriptions about the sensor. 

There are some detailed comments on this manuscript:

1. The title is "POF sensor for detecting and monitoring leakage current in a 500-kV transmission line along with weather conditions", but there are very few descriptions about the sensor, such as the sensing principle,  packaging and installation details, and the basic measurement ability of the sensor. Why choose POF to monitor the leakage current. What are the detailed advantages and disadvanteges? What is the related research on this point? How to prove the method is novel?

2. The field test results are good, but what is the main novelty of the paper? The abstract and the conclusion are too redundancy to present the main novelty of the paper. The manuscript is more like a project report but not a journal publication.

3. In Fig. 9, the subfigures can be replaced by a frequency response curve which represent the frequency response scope, but not screenshots of oscilloscope for three cases.

4. What are the influencing factors on sensor measurement? How the calibration curves be influeced by these main factors? It must be discussed.

 5. More related references on the fiber sensors to monitor the leakage current must be included or supplemented.

6. English could be more concise.

7. The writing style and figures of the paper are not standard and could be improved.

Author Response

Please, see below the answers for each of your comments. They have been included in the new version of the manuscript (yellow letters).

1) The title is "POF sensor for detecting and monitoring leakage current in a 500-kV transmission line along with weather conditions", but there are very few descriptions about the sensor, such as the sensing principle,  packaging and installation details, and the basic measurement ability of the sensor. Why choose POF to monitor the leakage current. What are the detailed advantages and disadvanteges? What is the related research on this point? How to prove the method is novel?

ANSWER

The sensing principle is based on the ability of a high-brightness LED in providing an output light proportional to the current passing through it. The LED color wavelength chosen for this work provides the best temperature-independent output light, compromised with the optical fiber’s smallest attenuation window, 70 dB/km at 565 nm. For guiding the LED light to the receptor at the ground potential, a 50-m plastic optical fiber (POF) cable was used, that presents at the chosen transmission window a total attenuation of 3.5 dB.

Advantages of POF as compared with silica fiber are the higher diameter that allows easier manipulation and connectoring, small bending losses due to its robust guiding capability with high numerical aperture, and super-multimode characteristics. Additionally, tools, devices, and accessories applied for POF are much cheaper than their counterparts for silica fibers. The disadvantage of POF as compared with silica fiber is the higher attenuation that compromises long-distance telemetry, in our case this high attenuation caused a 3.5-dB signal attenuation.

As for the proofing that there are novelties in this work, we can mention: In searching the scientific literature in several periodicals, particularly those of IEEE, we could not find equivalent outcomes as those listed in the conclusions, such as:  

a) Continuous monitoring for three years of leakage current together with three weather parameters.

b) For the first time, a robust and reliable TRL 8 device was installed on 500-kV towers to monitor and long-distance transmitting leakage currents and weather parameters.

c) Broadband response and acquisition capabilities of partial discharge pulses in high-voltage insulators, allowing detection of high-frequency pulses.

d) Partial discharge surges are caused when the local temperature reaches the dew point and not by the simple existence of high humidity, as stated in many works dealing with leakage currents.

2) The field test results are good, but what is the main novelty of the paper? The abstract and the conclusion are too redundancy to present the main novelty of the paper. The manuscript is more like a project report but not a journal publication.

ANSWER

The novelties were listed above. Abstract and Conclusion were improved.

3) In Fig. 9, the subfigures can be replaced by a frequency response curve which represent the frequency response scope, but not screenshots of oscilloscope for three cases.

ANSWER

Done

4) What are the influencing factors on sensor measurement? How the calibration curves be influeced by these main factors? It must be discussed.

ANSWER

The main factor that can influence the sensor performance is the LED aging effect that keeps decreasing the LED output power. This factor can be easily circumvented by a periodical calibration of the sensor and inserting a new calibration curve (Fig. 8) in the PLD.

Temperature, as shown in Fig. 7, can change the sensor’s response, but causes only a maximum error of 3.8%, being negligible given the qualitative characteristic of this measurement where the importance is to check whether leakage currents are stable or increasing.

Optical cable length causes attenuation of 0.07 dB/m. This factor, however, was circumvented by calibrating all sensors with the same length of POF cable of 50 m as in the calibration curve shown in Fig. 8.

5) More related references on the fiber sensors to monitor the leakage current must be included or supplemented.

ANSWER

Apart from the references already mentioned, we included three more; all of them present optical fiber sensing methodology, such as:

a) Fontana, S. C. Oliveira, F. J. M. M. Cavalcanti, R. B. Lima, J. F. Martins-Filho and E. Meneses-Pacheco, "Novel sensor system for leakage current detection on insulator strings of overhead transmission lines," in IEEE Transactions on Power Delivery, vol. 21, no. 4, pp. 2064-2070, Oct. 2006, doi: 10.1109/TPWRD.2006.877099.

b) J. Wang, C. Yao, Y. Mi, X. Zhang and C. Li, "Research for the LED Optical Fiber Sensor for the Leakage Current of the Insulator String," 2012 Asia-Pacific Power and Energy Engineering Conference, 2012, pp. 1-4, doi: 10.1109/APPEEC.2012.6307625.

c) Yan Mi, Shaoqin Rui, Shoulong Dong, Chenguo Yao, Chengxiang Li and Zhongyong Zhao, "Optical fiber sensor system for monitoring leakage current of post insulators based on RBF neural network," 2014 ICHVE International Conference on High Voltage Engineering and Application, 2014, pp. 1-4, doi: 10.1109/ICHVE.2014.7035513.

6) English could be more concise.

ANSWER

We removed redundancies in order to improve the text.

7) The writing style and figures of the paper are not standard and could be improved.

ANSWER

We improved figures and writing style.

Reviewer 3 Report

The submitted text presents a description of a measuring system dedicated to the monitoring of leakage current in the transmission lines. The text is written clearly and easy to understand. However, it lacks scientific content. Apart from that, in my opinion, the text is not appropriate for MDPI Sensor journal, the content and main findings are related to power engineering and the text should be submitted to a journal dedicated to power engineering. In my opinion, the text should be rejected. Despite this, I will provide some directions and comments for improving the text for authors.

1.    Content of the text, in my opinion, is out of the scope of sensors journal. It would be better to submit text to a journal related to power engineering or energetics, for example, MDPI Energies ISSN: 1996-1073 or MDPI Electricity ISSN: 2673-4826. Readers of these journals would be more interested in the authors' findings.
2.    The presented results are very interesting, but in my opinion, they are more of a description of the engineering work carried out. The text lacks an element of science. The authors' conclusions are trivial, known in advance and presented by the authors in the introduction – lines 458, 459 and 460 – ‘The main conclusion of this study is that polluted insulators with marine NaCl are likely to present LC peaks when wet, either from condensation due to the dew point being reached and also by sudden thunderstorms that help the surging of discharge arcs’.
3.    Title of the text could be better and could better reflect the content of the text and the actual work results. Abbreviation ‘POF’ is not widely used. It was the first time I had seen it used. It would be better to just use the name ‘optical fiber’. Mentioning in the title ‘500-kV’ is not needed. Described problem and proposed solution, in my opinion, are universal and would work, maybe after some modification with other transmission line voltages (eg. 110-kV). So to conclude, a shorter title without abbreviations will be more informative for the common reader.
4.    Term ‘large-band sensors’ (line 58) is not technically correct. The authors are highly inconsistent with used terms, not always using the appropriate word. In the case ‘large-band’ there is a good term ‘broadband’.
5.    Figure from 10 to Figure 13 would be more informative when data in figures would be extracted, and converted in plain graph.

Author Response

Please, see below the answers for each of your comments. They have been included in the new version of the manuscript (yellow letters).

1) Content of the text, in my opinion, is out of the scope of sensors journal. It would be better to submit text to a journal related to power engineering or energetics, for example, MDPI Energies ISSN: 1996-1073 or MDPI Electricity ISSN: 2673-4826. Readers of these journals would be more interested in the authors' findings.

ANSWER

Just noting that we have submitted this manuscript to the special issue “Recent Progress in Optical Voltage and Current Sensors", which we think embraces the subject of the present work.

2) The presented results are very interesting, but in my opinion, they are more of a description of the engineering work carried out. The text lacks an element of science. The authors' conclusions are trivial, known in advance and presented by the authors in the introduction – lines 458, 459 and 460 – ‘The main conclusion of this study is that polluted insulators with marine NaCl are likely to present LC peaks when wet, either from condensation due to the dew point being reached and also by sudden thunderstorms that help the surging of discharge arcs’

ANSWER

This paper, based on previous research, presents improvements such as a broadband receiver, a robust and reliable sensing and transmitting system, and the inclusion of dew point information. These improvements made it possible to detect high-frequency partial discharges, study the behavior of polluted insulators for a long period of time (three years), and correlate climate parameters with the occurrence of leakage currents. Since many conclusions not previously described were reached, the authors ponder that there are scientific elements that support the publication.

We agree with your view that the sentence on lines 458, 459, and 460 is not exactly a conclusion. In fact, we wrongly mixed a known fact with a concluded one. What is generally known is that when a polluted insulator gets wet, it conducts current (Wang et al., 2017). Our conclusion is that partial discharge surges are caused when the local temperature reaches the dew point and not by the simple existence of high humidity, as stated in many works dealing with leakage currents (e.g. Meyer et al., 2011).

3) Title of the text could be better and could better reflect the content of the text and the actual work results. Abbreviation ‘POF’ is not widely used. It was the first time I had seen it used. It would be better to just use the name ‘optical fiber’. Mentioning in the title ‘500-kV’ is not needed. Described problem and proposed solution, in my opinion, are universal and would work, maybe after some modification with other transmission line voltages (eg. 110-kV). So to conclude, a shorter title without abbreviations will be more informative for the common reader

ANSWER

Done.

4) Term ‘large-band sensors’ (line 58) is not technically correct. The authors are highly inconsistent with used terms, not always using the appropriate word. In the case ‘large-band’ there is a good term ‘broadband’.

ANSWER

Done.

5) Figure from 10 to Figure 13 would be more informative when data in figures would be extracted, and converted in plain graph.

ANSWER

Done.

Round 2

Reviewer 2 Report

Most concerns last time are answered by the authors. But there are still some points which are not clear to the readers.

1. The paper provides an important field test case for the designed fiber-optic current sensor. However, where are the different sensors installed? It is suggested that in Fig. 5, all the sensors used are marked clearly in the monitoring topology.

It is said there are three locations for sensing, but in the results, there is only result at one location during different periods. Then what's the difference among the three different monitoring locations? 

2. The English should be polished again.

Author Response

Question 1) About sensor's location and difference among monitoring locations: The sensors are indicated in Fig. 5 as Tower 2, Tower 13, and Tower 19 which are relatively close to each other. We did not observe different degrees of pollution when comparing data from different towers, which confirmed that the LT was relatively short and completely encompassed in the local micro-climate.

Question 2) About the English Language: We contacted a native English speaker who performed a review and made minor spell and typos corrections.